# The Effect of Decreased Antipseudomonal Drug Consumption on *Pseudomonas aeruginosa* Incidence and Antimicrobial Susceptibility Profiles over 9 Years in a Lebanese Tertiary Care Center

**DOI:** 10.3390/antibiotics12020192

**Published:** 2023-01-17

**Authors:** Rima El-Basst, Sanaa Saliba, Lama Saleh, Nicolas Saoud, Eid Azar, Pierre Zalloua, Amanda Chamieh

**Affiliations:** 1Department of Infectious Diseases, Saint George Hospital University Medical Center, Beirut P.O. Box 166378, Lebanon; 2Department of Infection Prevention and Control and Antimicrobial Stewardship, Saint George Hospital University Medical Center, Beirut P.O. Box 166378, Lebanon; 3College of Medicine and Health Sciences, Khalifa University, Abu Dhabi P.O. Box 127788, United Arab Emirates; 4Harvard T.H. Chan School of Public Health, Boston, MA 02115, USA

**Keywords:** *Pseudomonas aeruginosa*, antimicrobial resistance, consumption, carbapenems, stewardship, difficult-to-treat

## Abstract

*Pseudomonas aeruginosa* (PAE) is intrinsically resistant to numerous classes of antimicrobials such as tetracycline and β-lactam antibiotics. More epidemiological surveillance studies on the antimicrobial susceptibility profiles of PAE are needed to generate clinically significant data and better guided therapeutic options. We describe and analyze in a retrospective study the epidemiologic trends of 1827 *Pseudomonas* spp. isolates (83.5% PAE, 16.4% *Pseudomonas* sp., and 0.2% *Pseudomonas putida*) from various clinical specimens with their resistance patterns to antimicrobial consumption at a tertiary medical center in Lebanon between January 2010 and December 2018. We report a significant drop in the incidence of PAE from sputum (*p*-value = 0.05), whereas bloodstream infection isolation density showed no trend over the study period. We also registered a minimal but statistically significant drop in resistance of Pseudomonas to certain antibiotics and a decrease in the consumption of antipseudomonal antibiotics (*p*-value < 0.001). Only 61 PAE isolates from a total of 1827 Pseudomonas cultures (3.33%) were difficult to treat, of which only one was a bacteremia. Interestingly, we found that the carbapenem susceptibility of Pseudomonas was unaffected by the decrease in their consumption. These results augur that antimicrobial pressure may not be the sole contributor to resistance emergence. Finally, antimicrobial stewardship seems to have a positive impact on nosocomial epidemiology.

## 1. Introduction

Over the years, *Pseudomonas aeruginosa* has become notoriously implicated in community infections, as well as difficult-to-treat, severe, and life-threatening infections [1,2]. It is the most common etiology of Gram-negative bloodstream infections (BSI) among hospitalized patients and the third most common etiology of BSI in hospitalized and community-dwelling patients [3,4].

*PAE* is intrinsically resistant to numerous classes of antimicrobials such as tetracycline and β-lactam antibiotics. It can acquire resistance to commonly prescribed antimicrobials during treatment through a mechanism called selective pressure. Selective pressure is a bacterial mechanism where, under antibiotic presence, susceptible strains disappear, facilitating the survival of intrinsically resistant species [5]. Pseudomonas also have other mechanisms of resistance, such as horizontal transfer of resistance genes, changes in cell permeability, efflux or therapeutic target, and the selection of hypermutable clones. Resistance of PAE results in delays in adequate antimicrobial therapy and unfavorable clinical outcomes [5,6,7]. Patterns of antibiotic resistance in *PAE* change over time [8] and differ by type of infection, sometimes resulting in limited therapeutic options [9].

More epidemiological surveillance studies on antimicrobial susceptibility profiles of *PAE* are needed to generate clinically significant data to better guide therapeutic options [10]. In this study, we present a 9-year overview of antibiotic susceptibility patterns in *Pseudomonas species* isolates from various clinical specimens. We also demonstrate the impact of in-hospital antimicrobial consumption on these susceptibility patterns.

## 2. Results

### 2.1. General Results

A total of 1827 *Pseudomonas* spp. were isolated between January 2010 and December 2018 (642,564 total PD), of which 83.5% belonged to PAE and 16.4% belonged to *Pseudomonas* sp. Only three *Pseudomonas putida* were isolated over the study duration from sputum and urine cultures.

A total of 775 (42.4%) sputum cultures, 493 (26.9%) wound cultures, 360 (19.7%) urine cultures, 110 (6%) blood cultures, and 89 (4.8%) fluid cultures were isolated. There was no significant change or trend over the study period in the repartition of *Pseudomonas* spp. isolates among different specimen types (blood, sputum, urine, fluid, and wound) (Figure 1).

### 2.2. Pseudomonas Isolation Density and BSI Density

#### 2.2.1. *Pseudomonas aeruginosa*

*Pseudomonas aeruginosa* isolation density decreased steadily from 2.62/1000 PD in 2010 to 2.27/1000 PD in 2018 (Gradient per year: −6%). *PAE* isolated from sputum was highest in 2010, with 1.27/1000 PD, and dropped significantly over the years to reach its lowest level in 2018, with 0.74/1000 PD (Gradient per month: −4.9%, *p*-value = 0.05). Fluid cultures containing PAE increased from 0.09/1000 PD in 2010 to more than double in 2018, reaching 0.22/1000 PD (Sen’s slope = +0.02, *p*-value = 0.03). The changes in PAE isolation densities were not translated into a change in bloodstream infections (BSI), which maintained a practically constant rate over the study period with a mean of 0.13 ± 0.04/1000 PD (Figure 2).

The changes in isolation density from other cultures were not statistically significant.

#### 2.2.2. *Pseudomonas* sp.

In 2010, a 0.64/1000 PD rate was registered for the isolation density of *Pseudomonas* sp., which increased to 0.93/1000 PD in 2011 and then significantly dropped over the study period, reaching a minimal value of 0.16/1000 PD in 2018 (Sen’s slope = −0.07, *p*-value < 0.001).

Similarly, *Pseudomonas* sp. isolated from sputum, urine, and wound cultures decreased significantly from 2010 to 2018 with *p*-values of 0.01, 0.02, and 0.05, respectively.

The changes in isolation density from other cultures were not statistically significant.

#### 2.2.3. *Pseudomonas putida*

Only three *Pseudomonas putida* were isolated over the study period: one in 2014 from a urine culture, and two in 2017, one from urine and one from a sputum culture.

### 2.3. Resistance of Pseudomonas aeruginosa to Different Antibiotics

*Pseudomonas aeruginosa* isolates were tested for resistance to antipseudomonal antibiotics listed in the table below (Table 1). We found a statistically significant downward trend of resistance to cephalosporins (FEP and CAZ), aminoglycosides (GEN, AMK), and ofloxacin (*p* < 0.05 and R^2^ > 0.6) (Table 1).

#### 2.3.1. Difficult-to-Treat Cultures

A total of 61/64 difficult-to-treat cultures of Pseudomonas isolates were identified as PAE over 9 years of study, corresponding to 3.33% of all isolates. Only one isolate was from blood culture (0.05%). The origins of the other DTR isolates are shown in Table 2.

All these PAE DTR isolates still had therapeutic options, despite nephrotoxicity, whether aminoglycosides or polymyxins. One sputum culture belonging to *Pseudomonas* sp., isolated in 2010 in the ICU, was found to be resistant to all tested antibiotics, including AMK and COL.

#### 2.3.2. Cephalosporins

Resistance to FEP in 2010 was 22.2%, increased to reach its peak in 2011 (28.8%), and then decreased to reach its lowest rate in 2015 with 10.7%. In the last 3 years, resistance to FEP started increasing again from 11.4% in 2016 to 14.2% in 2018. Over the years, an overall downward trend of −2% per year was registered (*p*-value = 0.048, R^2^ = 0.66) (Figure 3a).

Similarly, resistance to CAZ decreased from 19.9% in 2010 to reach its lowest rate in 2016 with 9.5% resistant and increased in the last two years of the study, reaching 13.2% resistant and 12.6% resistant in 2017 and 2018, respectively. Mann−Kendall analysis showed an overall downward trend of −1% per year (*p*-value = 0.02, R^2^ = 0.6) (Figure 3a).

#### 2.3.3. Aminoglycosides

The highest yearly resistance to AMK was observed in 2011 (26.7%), and the lowest was observed in 2014 (12.0%), with an overall downward trend of −2% per year (*p*-value = 0.04, R^2^ = 0.69) (Figure 3b).

Resistance to GEN was at its highest in 2010 with 32.7% resistance; then, it started dropping to reach 16.8% resistant in 2018, with an overall downward trend of −1% per year (*p*-value = 0.006, R^2^ = 0.63).

#### 2.3.4. Quinolones

The highest yearly resistance to CIP was observed in 2012 with 36.5%. It dropped to its lowest in 2017 with 22.9% resistance, then increased to reach a resistance of 32.7%, a value close to what it was in 2010 (33.3% resistant). Mann−Kendall analysis showed no trend, with a mean resistance rate of 29.6% ± 5.2 (Figure 3c).

As for OFX, the highest resistance rate was in 2012, with 44.7%, and the lowest was in 2017, with 30.5%, with an overall downward trend of −2% per year (*p*-value = 0.03, R^2^ = 0.65) to reach 34.9% in 2018 (Figure 3c).

#### 2.3.5. Colistin

The yearly resistance rate for COL was highest in 2010 at 16.7%, then dropped to 0% in the years from 2012 to 2017, then reappeared in 2018 with 7.1% resistance. Mann−Kendall analysis showed no trend with a mean resistance rate of 4.8% ± 7.1.

In total, only three isolates out of 55 that were tested for COL were resistant; all three isolates were collected from the ICU. One culture (*Pseudomonas* sp.) isolated in 2010 was resistant to all tested antibiotics. The other two (*Pseudomonas* sp. and PAE) did not have a DTR profile and were isolated in 2011 and 2018, respectively.

#### 2.3.6. Carbapenems

Resistance to IPM showed no trend over the years, with a mean rate of 30.3% ± 3.3%. MEM was introduced in 2012 and showed the highest resistance rate of 50.0% that same year, then started dropping over the study period and registered a mean resistance rate of 33% ± 7.7%.

The resistance to both antipseudomonal carbapenems (360 isolates resistant to both IPM and MEM) and the resistances to each one of them individually (229 isolates resistant to either MEM or IPM) were determined. Resistance to antipseudomonal carbapenems was due to IMP with 26.9% of isolates being resistant in 2010, reaching 32.5% in 2012. After 2012, the number of isolates resistant to IPM only was almost equal to the number of isolates resistant to MEM only, with 3% and 2.5% resistant isolates, respectively. In 2014, resistance to MEM was higher than resistance to IPM (6.2% and 2.7%, respectively), after which a practically constant number for both with minor fluctuations was observed. After the introduction of MEM, resistance of *Pseudomonas aeruginosa* to antipseudomonal carbapenems was mainly combined (isolates were resistant to both IPM and MEM), with 38% resistant isolates in 2012, which decreased to 25% in 2013 and kept almost a constant rate (reaching 29% in 2018), while resistance to one of the antibiotics individually dropped from 32.5% in 2012 to 5.4% in 2013 and reached 3.6% in 2018 (Figure 4).

#### 2.3.7. Antipseudomonal Penicillin + β-Lactamase Inhibitors

Resistance to TZP showed no trend over the years, with a mean rate of 15.7 ± 5.6. It was highest in the first two years (22.2% and 24.6% in 2010 and 2011, respectively), then dropped to reach its lowest in 2015 and 2016 with 9.4% and 9.5%, respectively. In the last two years, however, the percentage of resistant isolates increased again, reaching 15.2% in 2018 (Figure 3d).

The highest yearly resistance of ATM was observed in the first years of the study (peak was in 2011 with 37.2% resistant), then dropped to reach its lowest value in 2016 with 10.5% resistant, with an overall downward trend of −2% (*p*-value = 0.059), registering 12.9% in 2018.

### 2.4. Antimicrobial Consumption

A total consumption rate of 423 DDD/1000 PD was observed for antipseudomonal antibiotics in January 2010. This consumption dropped to 263 DDD/1000 PD in December 2018. A significant (*p*-value < 0.001, R^2^ = 0.5) downward trend of −4.1% per month was observed over the study period (Table 3), with the highest consumption being in December 2013 (486 DDD/1000 PD) and the lowest being in January 2018 (161 DDD/1000 PD), representing a cumulative drop of 66.9%.

Imipenem consumption was highest in October 2010 with 110 DDD/1000 PD and dropped to its lowest in January 2018 with 3 DDD/1000 PD, showing a downward trend of −1.7% per month over the study period (*p*-value < 0.001, R^2^ = 0.91). The consumption rate of MEM was highest in October 2014 (100 DDD/1000 PD), then dropped again to 0 DDD/1000 PD in January 2018. Carbapenems consumption as a group decreased over the study period with a gradient of −1.7% per month (*p*-value < 0.001, R^2^ = 0.4).

The highest consumption rate of antipseudomonal quinolones was in January 2010 (179 DDD/1000 PD). It dropped over the study period at a mean rate of −0.4% per month, reaching 113 DDD/1000 PD in December 2018 (*p*-value = 0.01, R^2^ = 0.7) (Figure 3c).

The consumption of piperacillin−tazobactam fluctuated within a wide range but showed no trend over the study period, with an average rate of 30.8 ± 11.3 DDD/1000 PD (Figure 3d).

The mean consumption rate of cephalosporins was 26.9 ± 15.04 DDD/1000 PD. The Cefepime/Ceftazidime consumption rate varied over the months and registered a slight downward trend of −0.6% per month that was statistically significant (*p*-value < 0.001) but not representative (R^2^ = 0.21) (Figure 3a).

The changes in colistin and amikacin (Figure 3b) consumption rates were not statistically significant, with an average of 3.2 ± 2 DDD/1000 PD, and 7.05 ± 3.2 DDD/1000 PD, respectively.

## 3. Discussion

Our 9-year study on 1827 *Pseudomonas* spp. isolates, antipseudomonal drug consumption, and resistance at a tertiary care center in Beirut confirms that, despite the rise in multi-drug resistant (MDR), Gram-negative resistance in Lebanon and the surrounding region [11], Pseudomonas isolates decreased with a gradient of −6% per year from 2010 to 2018. This was mainly due to a significantly decreased isolation of *Pseudomonas* spp. in sputum. However, Pseudomonas isolates from blood cultures maintained a constant rate despite the changes in isolation density from other specimens. This shows that, although there may have been a decrease in colonization, Pseudomonas bloodstream infections (BSI) were not affected, and bacteremia rates maintained a stable value throughout the study period.

Several countries, including Lebanon, consistently showed high resistance levels for piperacillin−tazobactam, cephalosporins, carbapenems, monobactams, aminoglycosides, and fluoroquinolones [12]. While recent studies report a large increase (more than 50%) in antimicrobial resistance for *Pseudomonas aeruginosa* across many Arab countries [11] and Iran [13], in our study, the resistance profile patterns of Pseudomonas isolates suggest a significant downward resistance trend for cephalosporins (FEP and CAZ), aminoglycosides (GEN, AMK), and ofloxacin (*p* < 0.05 and R^2^ > 0.6). This has been previously demonstrated in recent studies for fluoroquinolones, especially in light of frequent safety warnings by drug regulatory agencies against their over prescription [14,15].

Over the study period, there was a significant drop in carbapenem consumption (*p*-value < 0.0001) due to the carbapenem-sparing strategy led by the antimicrobial stewardship program at SGHUMC since July 2016. Moreover, Pseudomonas’ carbapenem resistance had a mean rate of 30.3% ± 3.3% and 33% ± 7.7% for IPM and MEM, respectively, consistent with previous studies [16]. Interestingly, Pseudomonas’ carbapenem resistance was unaffected by the decrease in consumption of IPM, which is in line with the findings of other studies [16,17,18]. Although some studies report a correlation between hospital resistance of PAE and carbapenem consumption levels [19], our results echo the findings of Álvarez-Marín et al. (2021) showing that the correlation between carbapenem consumption and incidence density of carbapenem-resistant *Pseudomonas aeruginosa* is minimal [20]. Importantly, the decrease in consumption of antibiotics was noted to have a greater impact on resistance in the case of *Acinetobacter baumanii* [8,21,22,23]. All these findings show that even though antimicrobial selective pressure plays a role in antibiotic resistance [24,25], mechanisms of antimicrobial resistance of Pseudomonas are multifactorial (such as carrying resistance genes and other mechanisms) [9,13,26], with a variable role of antimicrobial selective pressure depending on the pathogen, which is akin to the results of other studies [7,9,18,21,27].

In treating Pseudomonas, the multidrug resistance (MDR) and extensive drug resistance (XDR) burden is very important since it significantly limits the therapeutic options for the patients [28], and according to a study conducted by Al-Orphaly et al. (2021), a high level of MDR *Pseudomonas aeruginosa* is observed in Eastern Mediterranean countries, with 64.5% MDR being registered in Lebanon [12]. Finding treatment options for patients with PAE infections has not been a big challenge [17]. This is evidenced in our study where, over a 9-year period, only 61 PAE isolates from a total of 1827 Pseudomonas cultures (3.33%) were DTR, of which only one was a bacteremia. Moreover, in our study, only one isolate was found to be resistant to all tested antibiotics, and all the other DTR isolates had alternative therapeutic options available, underscoring the fact that challenges in treating *Pseudomonas aeruginosa* are rare.

Limitations: With the available data, clonal outbreaks that could have resulted in the observed antimicrobial susceptibility profiles could not be ruled out. Furthermore, molecular genotyping data that are needed to comprehensively investigate the antimicrobial mechanism of resistance of Pseudomonas were not available. Finally, resistance to COL was not tested systematically, which may have affected the results of COL resistance.

Despite these limitations, this study included a large amount of data on Pseudomonas isolates over a long-time frame that were analyzed based on antimicrobial resistance and consumption. The highlighted differences from the studies could be attributed to many factors related to the studied populations, the assays used, the specimen types, and the presence of stewardship and surveillance programs [13,29].

The correlation between antibiotic consumption and resistance changes dynamically over time and depends on several factors and thus should not be viewed as a constant [8]. Therefore, factors that may play a modulating role between antibiotic consumption and resistance among PAE isolates in Lebanon need to be further investigated.

## 4. Materials and Methods

This is a retrospective study conducted at Saint George Hospital University Medical Center (SGHUMC), a 333-bed tertiary care center in Beirut, Lebanon. We recovered clinical microbiology isolates, antimicrobial consumption, and the number of patient-days (PD) for a period spanning from 1 January 2010 to 31 December 2018 from our electronic antimicrobial stewardship program database. No patients’ medical records were accessed.

This study was approved under the “exempt category” by the Ethics Committee of the University of Balamand/Saint George Hospital University Medical Center (IRB−REC/O1018−2011320).

At SGHUMC, traditional microbiology techniques are used for the isolation and identification of bacterial species. Further profiling is conducted using API gallery (https://www.biomerieux−usa.com/clinical/api accessed on 20 October 2022). Our analysis included all *Pseudomonas species* isolated from blood, sputum, urine, wound, and fluid cultures. Duplicate specimens belonging to the same patient with similar antimicrobial susceptibilities were excluded. Identified isolates included *Pseudomonas aeruginosa*, *Pseudomonas putida*, and *Pseudomonas* sp. Using the standard Kirby-bauer disc diffusion technique on Mueller–Hinton agar (bioMérieux, Marcy l’Etoile, France), isolates were tested for susceptibility against aminoglycosides (amikacin AMK, Gentamicin GEN), antipseudomonal carbapenems (imipenem IPM, meropenem MEM), antipseudomonal cephalosporins (ceftazidime CAZ, cefepime FEP), antipseudomonal fluoroquinolones (ciprofloxacin CIP, ofloxacin OFX), antipseudomonal penicillin + β-lactamase inhibitors (piperacillin−tazobactam TZP), and polymyxins (colistin COL). Antimicrobial susceptibility testing (AST) results were interpreted according to Clinical Laboratory Standards Institute (CLSI) guidelines [30]. Resistance percentages were calculated as the number of isolates resistant (R) or intermediate (I), divided by the total number of tested isolates for each antibiotic individually, and multiplied by 100. All microbiology cultures results were uploaded to the WHONET software.

Difficult-to-treat (DTR) isolates were defined as cultures resistant to carbapenems, beta-lactams, fluoroquinolones, and additional antimicrobials where applicable (i.e., piperacillin−tazobactam, aztreonam) [31]. DTR isolates were further categorized by collection site (intensive care unit (ICU) vs. regular ward).

Antibiotic consumption in Defined Daily Dose (DDD) was determined according to the World Health Organization Anatomical Therapeutic Chemical (ATC) index (https://www.whocc.no/atc_ddd_index/, accessed on 9 December 2022). Its value represents the monthly average consumption of an antimicrobial per year per 1000 PD. Total antipseudomonal consumption was determined as the sum of consumptions of all antipseudomonal antibiotics. Note that aztreonam is not available for use in Lebanon.

All antimicrobial consumption rates and isolation density per 1000 PD were derived. The percentages of isolates resistant to a certain antibiotic among all the cultures tested to this antibiotic were calculated. The Mann−Kendall trend test and homogeneity test [95% confidence interval (CI); *p* < 0.05] were applied on isolation densities of *Pseudomonas* spp., antimicrobial consumption, and antimicrobial susceptibility trends to analyze the progression over time. Analyses were performed using Microsoft Excel 2016 with the XLSTAT 2014 add-on [32].

## 5. Conclusions

Our study is the first work conducted over a 9-year period in Lebanon that describes and analyzes the epidemiologic trends of all *Pseudomonas* spp. isolates from various clinical specimens and their resistance patterns in relation to antimicrobial consumption. We report a significant drop in pseudomonal incidence, with only a few having limited therapeutic options, and an overall improvement in the resistance pattern over the study period. Antimicrobial stewardship seems to have a positive impact on nosocomial epidemiology. Our data serve as a pilot for further studies correlating the consumption and resistance of different antibiotics, their significance, and their interplay for a healthier hospital ecology.

## Figures and Tables

**Figure 1 antibiotics-12-00192-f001:**
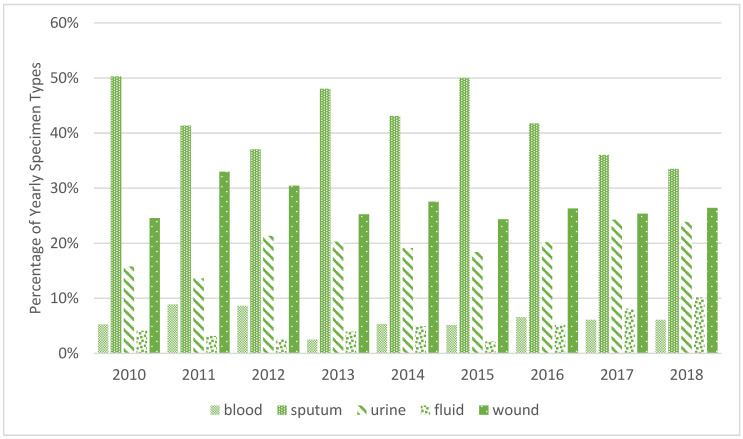
Yearly site distribution of *Pseudomonas* spp. isolates, in percentages.

**Figure 2 antibiotics-12-00192-f002:**
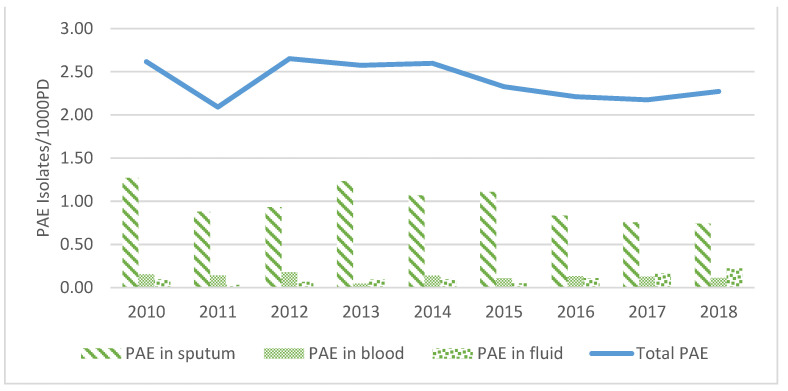
*Pseudomonas aeruginosa* trend from 2010 to 2018 by specimen type, per 1000 PD.

**Figure 3 antibiotics-12-00192-f003:**
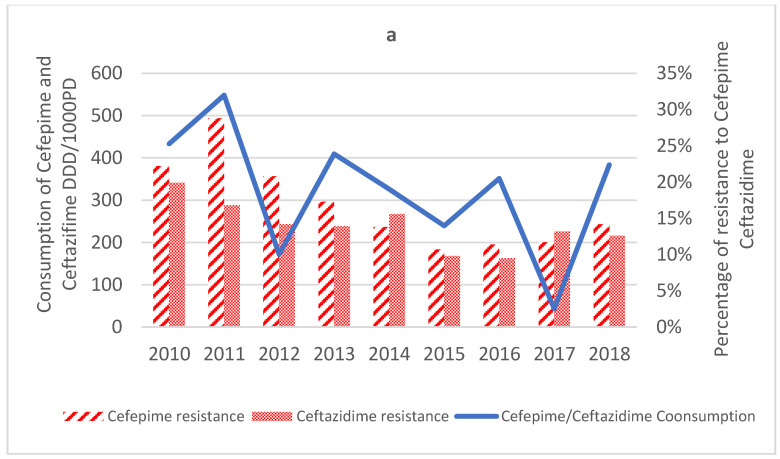
Consumption of antibiotics DDD/1000 PD vs. Percentage of PAE resistance to those antibiotics. (**a**) Consumption of Cefepime and Ceftazidime DDD/1000PD vs. Percentage of PAE resistance to Cefepime and Ceftazidime; (**b**) Consumption of Amikacin DDD/1000PD vs. Percentage of PAE resistance to Amikacin; (**c**) Consumption of Ciprofloxacin and Ofloxacin DDD/1000PD vs. Percentage of PAE resistance to Ciprofloxacin and Ofloxacin; (**d**) Consumption of Piperacillin–Tazobactam DDD/1000PD vs. Percentage of PAE resistance to Piperacillin–Tazobactam.

**Figure 4 antibiotics-12-00192-f004:**
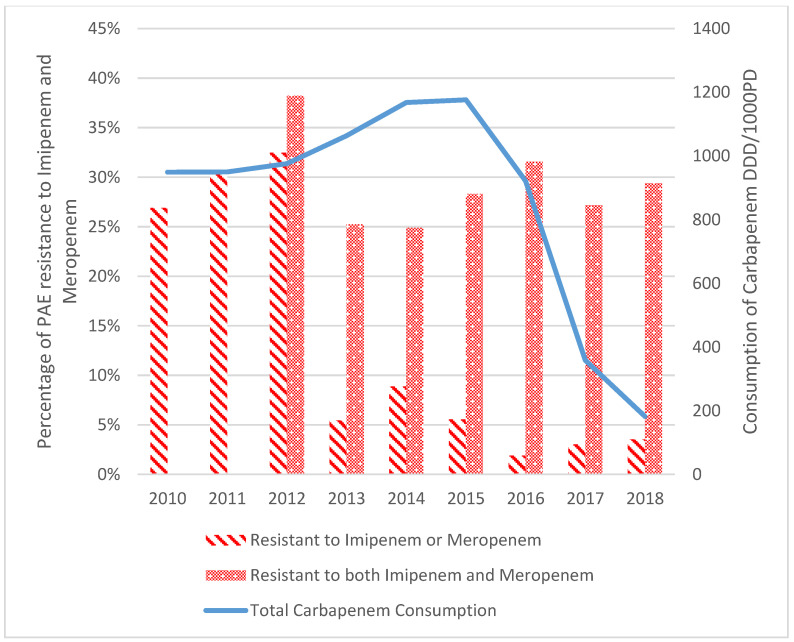
Change in consumption of carbapenems and resistance of *Pseudomonas aeruginosa* to MEM and IPM from 2010 to 2018.

**Table 1 antibiotics-12-00192-t001:** Trends of Pseudomonas resistance to antibiotics.

Antibiotic Resistance	Gradient	95% CI	R-Squared	*p*-Value	Trend
Cefepime (*n* = 1818)	−0.018	−0.03 to −0.01	0.662	0.048	↓
Ceftazidime (*n* = 1817)	−0.009	−0.02 to −0.003	0.557	0.016	↓
Ciprofloxacin (*n* = 1821)	−0.014	−0.03 to −0.001	0.377	0.076	↔
Colistin (*n* = 55)	0.000	−0.03 to 0.0	0.313	0.316	↔
Gentamicin (*n* = 1827)	−0.010	−0.03 to −0.004	0.629	0.006	↓
Imipenem (*n* = 1823)	0.001	−0.01 to 0.01	0.00003	0.834	↔
Meropenem (*n* = 1285)	−0.004	−0.04 to 0.01	0.303	0.548	↔
TZP (*n* = 1827)	−0.014	−0.03 to −0.003	0.561	0.076	↔
Aztreonam (*n* = 1796)	−0.022	−0.04 to −0.003	0.566	0.059	↔
Amikacin (*n* = 1827)	−0.015	−0.03 to −0.003	0.69	0.036	↓
Ofloxacin (*n* = 1802)	−0.0163	−0.02 to −0.01	0.646	0.029	↓

Values in bold are considered statistically significant (R−squared > 0.5 and *p*-value < 0.05). n indicates the number of tested isolates for the corresponding antibiotic. TZP: Piperacillin−tazobactam. ↓: downward trend;↔: no trend.

**Table 2 antibiotics-12-00192-t002:** Number of difficult-to-treat *P. aeruginosa* isolates per specimen per year.

Year	Blood	Sputum	Urine	Wound
2010	0	2	0	0
2011	0	8	0	2
2012	0	1	2	2
2013	0	4	2	1
2014	0	3	3	3
2015	0	3	1	1
2016	0	2	2	1
2017	0	2	2	2
2018	1	5	5	1
Total	1	30	17	13

**Table 3 antibiotics-12-00192-t003:** Trends of antibiotic consumption between 2010 and 2018.

Antibiotic Resistance	Gradient	95% CI	R−Squared	*p*-Value	Trend
Cephalosporins	−0.006	−0.01 to −0.003	0.21	**0.000**	↔
Quinolones	−0.004	−0.01 to −0.001	**0.702**	**0.013**	↓
Colistin	0.000	0 to 0.001	0.084	0.029	↔
Imipenem	−0.017	−0.020 to −0.015	**0.908**	**<0.0001**	↓
Meropenem	0.001	−0.01 to 0.01	0.00006	0.764	↔
Carbapenems	−0.017	−0.02 to −0.01	0.399	**<0.0001**	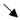 *
Piperacillin−tazobactam	0.000	−0.003 to 0.003	0.013	0.926	↔
Amikacin	0.001	0.0 to 0.001	0.067	0.079	↔
All Anti-pseudomonal antibiotics	−0.0407	−0.05 to −0.03	0.485	**<0.0001**	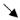 *

Values in bold are considered statistically significant (R-squared > 0.5 and *p*-value < 0.05). ↓: downward trend; ↔: no trend; * downward trend with only *p*-value < 0.05.

## Data Availability

All data are included in the manuscript.

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
