# Peer review of "The Effect of Decreased Antipseudomonal Drug Consumption on Pseudomonas aeruginosa Incidence and Antimicrobial Susceptibility Profiles over 9 Years in a Lebanese Tertiary Care Center"

_antibiotics, 2023, doi:10.3390/antibiotics12020192_

Round 1

Reviewer 1 Report

The author's approach in this paper is the incidence of various pseudomonas in different communities over 9 years. Along with these data, they related the resistance rate of different antibiotics in the same years. In my opinion, this article lacks some necessities. It has only 33 references and during the discussion few works that corroborate the data obtained. They do not establish a clear relationship between the resistance/susceptibility rate of different antibiotics and the increase/decrease of different resistances. During the results, they only intensely described the data obtained, they did not regret anything relevant.

Major Revision

Line 33- "Over the years, Pseudomonas aeruginosa has become notoriously implicated in community infections as well as difficult-to-treat, severe, life-threatening infections. [1], [2] It is the most common etiology of gram-negative bloodstream infections (BSI) among hospitalized patients and the third most common etiology of BSI in hospitalized and community-dwelling patients [3,4]" - Throughout this work, the opposite was demonstrated, that the infection rate in blood infections was quite low. How to explain what happened? Urinary infections have a high prevalence of P. aeruginosa. Check : doi: 10.1007/s00284-011-0026-y , doi.org/10.2147/IDR.S346313  etc

Line 40 - "Selective pressure is a bacterial mechanism where under antibiotic presence susceptible strains disappear, facilitating the survival of intrinsically resistant species...." - The biofilm makes a significant contribution to this strain selection. Pseudomonas aeruginosa is one of the largest producers of biofilms, a paragraph talking about this topic would complete the article. You can check this information at doi: 10.3390/ijms222312892.

Line 217 - "Several countries including Lebanon consistently showed high resistance levels for piperacillin-tazobactam, cephalosporins, carbapenems, monobactams, aminoglycosides, and fluoroquinolones" - Wasn't that to be expected, since most of these antibiotics are β-lactams? During the introduction, the authors wrote "P. aeruginosa is intrinsically resistant to numerous classes of antimicrobials such as tetracycline and β-lactam antibiotics", so nothing relevant adds to this study. 

Line 221 to 226 - The phrase makes no sense, they compare antibiotics from other classes with fluoroquinolones. They must compare each antibiotic with its class. 

The authors correlate data from IPM and MEM antibiotics alone and together. What is the reason for the data discrepancy? Together they worked as synergists and the acquisition of resistance is greater? Since they are of the same class, being resistant to one, wasn't it expected to be resistant to the other (since the mechanism is the same)?

Is the increase in the use of antibiotics consistent with the number of resistances they have obtained?

Colistin conclusions are too vague. If the authors state that colistin has not been systematically tested, shouldn't they delete that data or take a different approach?

Minor revision

Line 17, 29, 45, 52, 57 etc - Pseudomonas aeruginosa must be in italics. Check for all text.

Line 17, 58, 59 etc - Pseudomonas Putida e Pseudomonas spp.  must be in italics. Check for all text.

Line 65 - Figure 1 has low quality.

Line 68 -The subtitle name 2.2.1 is the same as 2.2.2.

Line 120 -Figure 2  has low quality.

 Line 138 - "Polymyxins (Colistin)" - Polymyxins can be divided into polymyxin E, polymyxin B etc. If you speak only of colistin, you must mention polymyxin E.

Author Response

We would like to thank the reviewers for their time and effort in helping us improve the quality of our work. It shows that our manuscript was read and reviewed meticulously. We have answered all requests and amended what is possible.

Reviewer 1:

The author's approach in this paper is the incidence of various pseudomonas in different communities over 9 years. Along with these data, they related the resistance rate of different antibiotics in the same years. In my opinion, this article lacks some necessities. It has only 33 references and during the discussion few works that corroborate the data obtained. They do not establish a clear relationship between the resistance/susceptibility rate of different antibiotics and the increase/decrease of different resistances. During the results, they only intensely described the data obtained, they did not regret anything relevant.

We report the incidence of various pseudomonas isolates in one community which is Saint George Hospital University Medical Center (SGHUMC) over 9 years, as we indicate in our methods. We add to this the rate of anti-pseudomonal antimicrobial consumption at SGHUMC in the same time period. The issue of linking antimicrobial consumption to observed antimicrobial resistance/susceptibility rates is multifactorial and cannot be taken as a direct causality. Moreover, the relationship between antimicrobial consumption and observed resistance is pathogen dependent. For example, a phenomena observed on Acinetobacter baumannii may not be true for  Pseudomonas sp., as we have seen in previous studies.

Major Revision
Line 33- "Over the years, Pseudomonas aeruginosa has become notoriously implicated in community infections as well as difficult-to-treat, severe, life-threatening infections. [1], [2] It is the most common etiology of gram-negative bloodstream infections (BSI) among hospitalized patients and the third most common etiology of BSI in hospitalized and community-dwelling patients [3,4]" - Throughout this work, the opposite was demonstrated, that the infection rate in blood infections was quite low. How to explain what happened? Urinary infections have a high prevalence of P. aeruginosa. Check : doi: 10.1007/s00284-011-0026-y , doi.org/10.2147/IDR.S346313  etc

This reflects the local epidemiology observed at SGHUMC which may contradict existing literature. The difference in the rate of isolation in urinary, respiratory, wound etc may reflect colonization at times vs true infection. However, isolation of P. aeruginosa in the bloodstream always indicates an infection. Thus, the true invasiveness of P. aeruginosa can be determined from the rate of observed bacteremia as opposed to in-depth chart review by multiple reviewers to determine infection vs colonization of other specimen types.  We demonstrate that the rate of bloodstream infections remains more or less constant over the years.  This shows the relative independence of Pseudomonas invasive disease despite changes in external factors such as colonization.

Line 40 - "Selective pressure is a bacterial mechanism where under antibiotic presence susceptible strains disappear, facilitating the survival of intrinsically resistant species...." - The biofilm makes a significant contribution to this strain selection. Pseudomonas aeruginosa is one of the largest producers of biofilms, a paragraph talking about this topic would complete the article. You can check this information at doi: 10.3390/ijms222312892.

Biofilm formation and its role in development of antimicrobial resistance, selective pressure and subsequent infection is a very complex and broad topic. To properly attribute biofilm to strain selection requires in-depth, detailed molecular analysis. Its discussion is beyond the scope of this manuscript which is aimed at describing the epidemiological trend of all pseudomonal isolates at our institute.

Line 217 - "Several countries including Lebanon consistently showed high resistance levels for piperacillin-tazobactam, cephalosporins, carbapenems, monobactams, aminoglycosides, and fluoroquinolones" - Wasn't that to be expected, since most of these antibiotics are β-lactams? During the introduction, the authors wrote "P. aeruginosa is intrinsically resistant to numerous classes of antimicrobials such as tetracycline and β-lactam antibiotics", so nothing relevant adds to this study. 

Our work demonstrated a decrease in the rates of resistance as opposed to other studies that indicate it is on the rise (line 219-223). In reference to line 217, we confirm that our local, single-center data is compatible with national data . The references in the discussion establish a relation with our obtained results.

Line 221 to 226 - The phrase makes no sense, they compare antibiotics from other classes with fluoroquinolones. They must compare each antibiotic with its class. 

These lines have been rephrased for clarity.

The authors correlate data from IPM and MEM antibiotics alone and together. What is the reason for the data discrepancy? Together they worked as synergists and the acquisition of resistance is greater? Since they are of the same class, being resistant to one, wasn't it expected to be resistant to the other (since the mechanism is the same)?

The mechanisms of resistance to imipenem and meropenem are different in pseudomonas that is why they are discussed separately. Meletis G, Exindari M, Vavatsi N, Sofianou D, Diza E. Mechanisms responsible for the emergence of carbapenem resistance in Pseudomonas aeruginosa. Hippokratia. 2012 Oct;16(4):303-7. PMID: 23935307; PMCID: PMC3738602.

Is the increase in the use of antibiotics consistent with the number of resistances they have obtained?

This is not applicable for all antibiotics and is discussed in the manuscript

Colistin conclusions are too vague. If the authors state that colistin has not been systematically tested, shouldn't they delete that data or take a different approach?

This is mentioned in the limitations

Minor revision

Line 17, 29, 45, 52, 57 etc - Pseudomonas aeruginosa must be in italics. Check for all text.

Line 17, 58, 59 etc - Pseudomonas Putida e Pseudomonas spp.  must be in italics. Check for all text.

Line 65 - Figure 1 has low quality.

Line 68 -The subtitle name 2.2.1 is the same as 2.2.2.

Line 120 -Figure 2  has low quality.

 Line 138 - "Polymyxins (Colistin)" - Polymyxins can be divided into polymyxin E, polymyxin B etc. If you speak only of colistin, you must mention polymyxin E.

All these requested revisions have been addressed and amended.

Reviewer 2 Report

General comments

This paper adds to the scientific and clinical data on Pseudomonas spp. and changes to resistance overtime. There this paper describes the change in trends of Pseudomonas spp. from key clinical specimens. It then reports the change in both the resistance of isolates to antipseudomonal antibiotics and antibiotic consumption over this time period. While overall a decrease in both the consumption and resistance to antibiotics was observed, the breakdown of each of the antibiotics and specimen type better reflect actual trends.

Many of my suggestions focus on clarifying this data and clearly defining terms.

1.       Needs to be made very clear when it is just pseudomonas sp vs Pseudomonas aeruginosa….this is not always clear

2.       Citations in text, move full-stop/period to outside the brackets

a.       [2,3]. Instead of . [2,3]

b.       This is the same when table or figure is referenced

3.       References

a.       Italicise bacteria names and use upper and lower case correctly

b.       Ie. Pseudomonas aeruginosa not Pseudomonas Aeruginosa

c.       Please check reference layout is in accordance with journal

                                                               i.      Eg. Reference number 1 needs reformatting

4.       Italics

a.       Throughout the body of the text and in the title bacteria names are not consistently italicised

                                                               i.      Please fix

5.       Abbreviations

a.       Please be consistent with how names are abbreviated

                                                               i.      Pseudomonas aeruginosa is abbreviated to both PAE and P. aeruginosa

                                                             ii.      When using abbreviations in figures/tables please make sure they are also written in full in footnote, especially as these abbreviations are not in the text at all

6.       Focus of manuscript:

a.       Is there a need for P. putida to be included? There are only 3 isolates isolated over the 9 years and no antimicrobial resistance data is included. Not sure these results add to the study. Additionally, as per Line 280 and 281 this is included in the Pseudomonas species. data. I think indicate the number and type of P. putida infections in a broader definition of Pseudomonas species and then remove them from the manuscript.

b.       Pseudomonas sp. versus Pseudomonas spp.

                                                               i.      When referring to the whole group of pseudomonas isolates represented in this work please use Pseudomonas spp. You include more than one individual species in this analysis and therefore the plural of species should be used.

                                                             ii.      Example Line 56. This should be ‘A total of 1827 Pseudomonas spp. were isolated…’.

Title

This manuscript describes the antibiotic trends of P. aeruginosa and Pseudomonas sp. (and includes P. putida), should the title better reflect this and replace ‘Pseudomonas aeruginosa’ with ‘Pseudomonas species’?

Abstract

·       Line 14: should the word ‘antibiotics’ be placed at the end of this sentence to read: ‘like tetracycline and ß-lactam antibiotics’.

·       Line 27: antimicrobial does not need to be capitalised

Introduction

·       Line 34: please add an ‘and’ in between the words severe and life-threatening

·       Line 52: should P. aeruginosa be replaced with Pseudomonas species?

Materials and Methods

·       How are the isolates identified? This should be included.

·       Why are only P. aeruginosa and P. putida successfully/formally identified and no other species?

·       If there are more than one type of Pseudomonas species included in the ‘Pseudomonas sp.’ (line 281 for example) this should be changed to ‘Pseudomonas spp.’ to better represent multiple species.

·       Line 273: What does ASP stand for? Please spell out in full

·       Line 299,300: ‘patient days’ does not need to be spelled out in full, the abbreviation can be used here. Previously defined on Line 272.

·       I think a section (whether it is here or in results) focusing on how many isolates were screened for each antibiotic is important. I think the isolates which are not screened need to be removed from the analysis. Did you do this? This is not clear from the methods. Using a yearly isolate total rather than the number of isolates were there is data for each antibiotic will change the results but will be more representative of changes to resistance.

Results

·       Line 57: Patient-days can be abbreviated to PD

·       Line 58: Replace pseudomonas sp. with Pseudomonas spp. if more than one species is included

·       Line 64: Please move the period/full-stop to the outside of the Figure 1 brackets

·       Line 68: 2.2.1 Pseudomonas sp. – should this be Pseudomonas aeruginosa?

·       Figure 2:

1.       Label x axis

2.       Label both y axis

3.       Describe abbreviations or not use abbreviations in this figure

4.       I find this figure difficult to interpret. It needs to be made clear that the bars are using the left hand side y axis and the lines are using the right hand side axis.

5.       Why include BSI results and not wound or urine? You report they all have non-significant changes over the study period.

·       2.2.3 Pseudomonas putida: I think you could mention that this data is included in section 2.2.2 Pseudomonas sp. (change to spp.). If you do this, could this section be removed?

·       2.3 Resistance of Pseudomonas aeruginosa to different antibiotics: Could you include the denominator here…how many isolates were tested for each of the antibiotics? This would add to understanding these results. Especially considering the colistin section.

·       Line 96: Change CZ to CAZ

·       Line 96: ATM, this is not in the antibiotic list in the methods, is this azithromycin? If so, this needs to be included in the antibiotic listing in the methods.

·       Table 1: Should the title be Pseudomonas aeruginosa rather than Pseudomonas?

·       Line 106: Please replace P. sp with Pseudomonas spp.

·       Table 2: Should the title be Pseudomonas spp. rather than P. aeruginosa?

·       Line 142: Were only 55 of the 1827 isolates tested for colistin resistance? If so, how can you report trends? Surely there is insufficient results for each of the years. Is that why there is 0% from 2012 to 2017? No isolates were tested? If this is the case colistin resistance needs to be reported differently.

·       Lines 143 and 144: Please be consistent with abbreviations here. Please follow what is used throughout the manuscript

·       Figure 3, General comments:

·       Does this include all Pseudomonas isolates? If so, the title should read ‘Pseudomonas spp.’

·       Is very difficult to read. It is very small and the font is almost illegible. I would suggest fixing before publication.

·       I think the figure should be set out in the order you read the text in. Aminoglycosides come second in the text but are last (3d) in the figure

·       Please label all the axis. Please make it clear that the bars or consumption are using the left hand side and the resistance are using right hand side of the y axis?

·       How the axes are labelled is a little misleading. For example, in 3c you mention in the text that in 2015 & 2016 the rates were 9.4 % and 9.5%. on the graph the orange line is sitting above the 10% value. I think these lines need to be correctly marked and the data graphed correctly. It could be an artifact of the size of the graph and can be easily fixed

·       Figure 3, 1a, Cephalosporins, this should be 3a

·       Figure 3, 3d, Amikacin, where are the gentamycin results and if added in please change the title to aminoglycosides

·       Lines 188, 189: This is misleading as meropenem was not introduced until 2012. Therefore, you cannot include 2010 and 2011 into these results. Has the statistics be analysed for years were meropenem was not being administered. This either needs to be changes or a more thorough explanation of the data needs to be included.

Discussion

·       Line 211: Can you be clearer with this sentence. Are you referring to a decline in the number of sputum samples or a decline in the number of pseudomonas sp. which are resistant isolated from sputum samples? If it is a decline in the number of sputum samples then this contradicts Line 62 – 64 where you mention no change in the number of sputum samples collected over the study period.

·       Line 212: extra comma that needs to be deleted

·       Line 225: You mention that stewardship programs are monitoring and changing prescription behaviours based on recommendations (Line 229). Has a similar recommendation been introduced for fluoroquinolones in light of safety warnings? Could this be a reason for a decline in consumption? Is it being prescribed less due to changes in regulations? If so, when was this implemented? Can you tell this from consumption data?

·       Line 244: Spell out XDR as this is the first, and only, time you use it

·       Has or will this study changed stewardship programs at SGHUMC? If so, can you discuss this. What are the implications from this work?

·       Does consumption match prescription? Do you know, or is there a way to track what was prescribed to what was consumed? I know you mention you didn’t go through patient records but that would be interesting maybe for in-house records to see the correlation between each of the markers, resistance, consumption and prescription. Could this be included as a limitation?

·       Despite excluding duplicate specimens with similar susceptibility patterns, did you record previous infections and treatments? Do you know if previous infections and whether they responded to treatment influenced the clinician’s decision for current treatment approach. Also, previous treatment could impact current resistance patterns. Could you comment on this?

·        Is there any reason you think might be responsible for a decrease in the prevalence of Pseudomonas despite sampling remaining constant? Has there been a change to sample processing, microbiological techniques in the lab, has there been a change to patient population? Can you comment on the decline in the isolation of pseudomonas?

Limitations

A follow up study focusing on the prevalence of clonal strains and whether there are common resistance markers in these isolates would be very beneficial to further understand this work. I understand the limitations of undertaking such a study though and why it might not be feasible.

Author Response

We would like to thank the reviewers for their time and effort in helping us improve the quality of our work. It shows that our manuscript was read and reviewed meticulously. We have answered all requests and amended what is possible

Reviewer 2:

General comments Amended

This paper adds to the scientific and clinical data on Pseudomonas spp. and changes to resistance overtime. There this paper describes the change in trends of Pseudomonas spp. from key clinical specimens. It then reports the change in both the resistance of isolates to antipseudomonal antibiotics and antibiotic consumption over this time period. While overall a decrease in both the consumption and resistance to antibiotics was observed, the breakdown of each of the antibiotics and specimen type better reflect actual trends.

Many of my suggestions focus on clarifying this data and clearly defining terms.

  1. Needs to be made very clear when it is just pseudomonas sp vs Pseudomonas aeruginosa….this is not always clear
  2. Citations in text, move full-stop/period to outside the brackets       a.[2,3]. Instead of . [2,3]                                                                      b.This is the same when table or figure is referenced
  3. References                                                                                              a.Italicise bacteria names and use upper and lower case correctly  b.Ie. Pseudomonas aeruginosanot Pseudomonas Aeruginosa        c.Please check reference layout is in accordance with journal               i.Eg. Reference number 1 needs reformatting
  4. Italics                                                                                               a.Throughout the body of the text and in the title bacteria names are not consistently italicised                                                                  i.Please fix
  5. Abbreviations                                                                                   a.Please be consistent with how names are abbreviated            i.Pseudomonas aeruginosa is abbreviated to both PAE and P. aeruginosa ii.When using abbreviations in figures/tables please make sure they are also written in full in footnote, especially as these abbreviations are not in the text at all
  6. Focus of manuscript:                                                                                 a. Is there a need for P. putidato be included? There are only 3 isolates isolated over the 9 years and no antimicrobial resistance data is included. Not sure these results add to the study. Additionally, as per Line 280 and 281 this is included in the Pseudomonas species. data. I think indicate the number and type of P. putida infections in a broader definition of Pseudomonas species and then remove them from the manuscript.

In fact, P. putida are only mentioned in the description of obtained isolates, all further analysis does not include them as you have suggested. We will make sure to clarify this further in the manuscript.

      b. Pseudomonas sp. versus Pseudomonas spp.                                             i.When referring to the whole group of pseudomonas isolates represented in this work please use Pseudomonas spp. You include more than one individual species in this analysis and therefore the plural of species should be used.

ii.Example Line 56. This should be ‘A total of 1827 Pseudomonas spp. were isolated…’.

Title

This manuscript describes the antibiotic trends of P. aeruginosa and Pseudomonas sp. (and includes P. putida), should the title better reflect this and replace ‘Pseudomonas aeruginosa’ with ‘Pseudomonas species’?

Our main results and majority of isolates are P. aeruginosa, which is why we mention it in the title. If you still believe we need to amend it please let us know.

Abstract Amended

  • Line 14: should the word ‘antibiotics’ be placed at the end of this sentence to read: ‘like tetracycline and ß-lactam antibiotics’.
  • Line 27: antimicrobial does not need to be capitalised

Introduction Amended

  • Line 34: please add an ‘and’ in between the words severe and life-threatening
  • Line 52: should P. aeruginosabe replaced with Pseudomonas species?

Materials and Methods

  • How are the isolates identified? This should be included. Amended
  • Why are only P. aeruginosa and P. putida successfully/formally identified and no other species?

At the time of the study, these were the isolates identified at our basic microbiology lab ,  using API gallery (https://www.biomerieux-usa.com/clinical/api). Other unidentified by this technology were categorized under P. sp.

  • If there are more than one type of Pseudomonas species included in the ‘Pseudomonas sp.’ (line 281 for example) this should be changed to ‘Pseudomonas spp.’ to better represent multiple species.
  • Line 273: What does ASP stand for? Please spell out in full

ASP: antimicrobial stewardship program

  • Line 299,300: ‘patient days’ does not need to be spelled out in full, the abbreviation can be used here. Previously defined on Line 272.
  • I think a section (whether it is here or in results) focusing on how many isolates were screened for each antibiotic is important. I think the isolates which are not screened need to be removed from the analysis. Did you do this? This is not clear from the methods. Using a yearly isolate total rather than the number of isolates were there is data for each antibiotic will change the results but will be more representative of changes to resistance.

We will make sure to clarify this in the methods since our analysis included different denominators for each antibiotic according to number of isolates tested/screened.

Results

  • Line 57: Patient-days can be abbreviated to PD
  • Line 58: Replace pseudomonas sp. with Pseudomonas spp. if more than one species is included
  • Line 64: Please move the period/full-stop to the outside of the Figure 1 brackets
  • Line 68: 2.2.1 Pseudomonas sp. – should this be Pseudomonas aeruginosa?
  • Figure 2:
  1. Label x axis
  2. Label both y axis
  3. Describe abbreviations or not use abbreviations in this figure
  4. I find this figure difficult to interpret. It needs to be made clear that the bars are using the left hand side y axis and the lines are using the right hand side axis.
  5. Why include BSI results and not wound or urine? You report they all have non-significant changes over the study period.

Despite the lack of statistically significant change in BSI trend, we included it in the graph because a BSI indicates true invasive infection vs possible colonization in wound/urine. It adds clinical relevance to the data. We will make sure to clarify the figures.

  • 2.2.3 Pseudomonas putida: I think you could mention that this data is included in section 2.2.2 Pseudomonas sp. (change to spp.). If you do this, could this section be removed?
  • 2.3 Resistance of Pseudomonas aeruginosato different antibiotics: Could you include the denominator here…how many isolates were tested for each of the antibiotics? This would add to understanding these results. Especially considering the colistin section.

This has been adjusted and N tested is added to table 1

  • Line 96: Change CZ to CAZ
  • Line 96: ATM, this is not in the antibiotic list in the methods, is this azithromycin? If so, this needs to be included in the antibiotic listing in the methods.

This refers to aztreonam, we will adjust it

  • Table 1: Should the title be Pseudomonas aeruginosarather than Pseudomonas?
  • Line 106: Please replace P. sp with Pseudomonas spp.
  • Table 2: Should the title be Pseudomonas spp. rather than P. aeruginosa?
  • Line 142: Were only 55 of the 1827 isolates tested for colistin resistance? If so, how can you report trends? Surely there is insufficient results for each of the years. Is that why there is 0% from 2012 to 2017? No isolates were tested? If this is the case colistin resistance needs to be reported differently.

As we mentioned in the manuscript, colistin is not systematically tested, usually, when faced with MDR Pseudomonas isolate, we request an additional testing for colistin. We will change the reporting of colistin testing according to your suggestion and the number of tested isolates was added as mentioned above.

  • Lines 143 and 144: Please be consistent with abbreviations here. Please follow what is used throughout the manuscript
  • Figure 3, General comments: Amended
  • Does this include all Pseudomonas isolates? If so, the title should read ‘Pseudomonas spp.’
  • Is very difficult to read. It is very small and the font is almost illegible. I would suggest fixing before publication.
  • I think the figure should be set out in the order you read the text in. Aminoglycosides come second in the text but are last (3d) in the figure
  • Please label all the axis. Please make it clear that the bars or consumption are using the left hand side and the resistance are using right hand side of the y axis?
  • How the axes are labelled is a little misleading. For example, in 3c you mention in the text that in 2015 & 2016 the rates were 9.4 % and 9.5%. on the graph the orange line is sitting above the 10% value. I think these lines need to be correctly marked and the data graphed correctly. It could be an artifact of the size of the graph and can be easily fixed

  • Figure 3, 1a, Cephalosporins, this should be 3a
  • Figure 3, 3d, Amikacin, where are the gentamycin results and if added in please change the title to aminoglycosides.

Only amikacin was included since gentamicin prescription and consumption is very low at SGHUMC.

  • Lines 188, 189: This is misleading as meropenem was not introduced until 2012. Therefore, you cannot include 2010 and 2011 into these results. Has the statistics be analysed for years were meropenem was not being administered. This either needs to be changes or a more thorough explanation of the data needs to be included.

We will clarify this in the manuscript, the analysis of meropenem trends was performed starting 2012.

Discussion

  • Line 211: Can you be clearer with this sentence. Are you referring to a decline in the number of sputum samples or a decline in the number of pseudomonas sp. which are resistant isolated from sputum samples? If it is a decline in the number of sputum samples then this contradicts Line 62 – 64 where you mention no change in the number of sputum samples collected over the study period.

We will clarify this in the manuscript.

  • Line 212: extra comma that needs to be deleted
  • Line 225: You mention that stewardship programs are monitoring and changing prescription behaviours based on recommendations (Line 229). Has a similar recommendation been introduced for fluoroquinolones in light of safety warnings? Could this be a reason for a decline in consumption? Is it being prescribed less due to changes in regulations? If so, when was this implemented? Can you tell this from consumption data?

The antimicrobial stewardship program strictly regulates carbapenems, piperacillin tazobactam, cefepime, ceftazidime but does not control fluoroquinolone. There has been no change in fluoroquinolone regulations in Lebanon. This may either indicate prescriber behavior changes vs simply a decline in the need to treat with fluoroquinolones.

  • Line 244: Spell out XDR as this is the first, and only, time you use it
  • Has or will this study changed stewardship programs at SGHUMC? If so, can you discuss this. What are the implications from this work?

This work led us to further understand that pseudomonas does not follow usual predictable patterns that we saw with other pathogens, it seems more or less unaffected by control of carbapenem consumption in isolate density and susceptibility. This led us to put in order stricter surveillance and molecular analysis of pseudomonal mechanisms of resistance and ST identification. We also expanded antimicrobial restricted list to include all cephalosporins and fluoroquinolones recently as a new antimicrobial stewardship interventions.

  • Does consumption match prescription? Do you know, or is there a way to track what was prescribed to what was consumed? I know you mention you didn’t go through patient records but that would be interesting maybe for in-house records to see the correlation between each of the markers, resistance, consumption and prescription. Could this be included as a limitation?

Prescription is tightly controlled by the antimicrobial stewardship program and infectious disease physicians that are part of it. Thus, the consumption numbers we have should be almost identical to prescription numbers. However, these numbers do not reflect individual differences that differ according to specific cases or patients. It may be interesting to start another prospective work assessing prescription vs consumption patterns and resistance at an individual level.

  • Despite excluding duplicate specimens with similar susceptibility patterns, did you record previous infections and treatments? Do you know if previous infections and whether they responded to treatment influenced the clinician’s decision for current treatment approach. Also, previous treatment could impact current resistance patterns. Could you comment on this?

We did not go through individual patient antimicrobial consumption, previous infection and treatments since that requires detailed chart review to determine. It is definitely interesting to look at this in a different, future work since antimicrobial consumption and link to development resistance is multifactorial and very complex, especially with a pathogen as versatile and adept as pseudomonas. We will amend this in the limitations section.

  • Is there any reason you think might be responsible for a decrease in the prevalence of Pseudomonas despite sampling remaining constant? Has there been a change to sample processing, microbiological techniques in the lab, has there been a change to patient population? Can you comment on the decline in the isolation of pseudomonas?

There have been no changes in microbiology techniques nor the patient population. We may conclude that the decline of pseudomonas may be the resolution of a pseudomonal outbreak that went unnoticed at the time. There may have been a dominating clone that for a reason or another disappeared, as is the case with most clones. Moreover, after the almost 80% decline in carbapenem consumption at SGHUMC, we were able to control Acinetobacter and Stenotrophomonas isolates starting in 2016. These changes may have also affected the balance and competition between different bacterial strains.

Limitations

A follow up study focusing on the prevalence of clonal strains and whether there are common resistance markers in these isolates would be very beneficial to further understand this work. I understand the limitations of undertaking such a study though and why it might not be feasible.

Round 2

Reviewer 1 Report

The article has been substantially improved. Congratulations!

Throughout the text there are some Pseudomonas words that are not in italics. In addition to standardizing the PAE, they either put it in italics or not, the text has both.